



# Effect of organic carbon addition on paddy soil organic carbon decomposition under different irrigation regimes

Heleen Deroo[1,2], Masuda Akter[3], Samuel Bodé[2], Orly Mendoza[1], Haichao Li[1], Pascal Boeckx[2], Steven Sleutel[1]

[1] Department of Environment, Ghent University, Ghent, Belgium

[2] Isotope Bioscience laboratory, Department of Green Chemistry and Technology,

   Ghent University, Ghent, Belgium

[3] Soil Science Division, Bangladesh Rice Research Institute, Gazipur, Bangladesh

*Correspondence to*: Heleen Deroo (Heleen.Deroo@UGent.be)

**Abstract.** Anaerobic decomposition of organic carbon (OC) in submerged rice paddies is coupled to the reduction of alternative soil electron acceptors, primarily $Fe^{3+}$. During reductive dissolution of $Fe^{3+}$ from pedogenic oxides, previously adsorbed native soil organic carbon (SOC) could be co-released into solution. Incorporation of crop residues could hence indirectly, i.e. through the stimulation of microbially mediated $Fe^{3+}$ reduction, promote the loss of native SOC via enhanced dissolution and subsequent mineralisation to $CO_2$ and $CH_4$. Our aim was to estimate the relevance of such a positive feedback

during the degradation of added OC, and to investigate the impact of irrigation management on this mechanism and on priming effects on native SOC decomposition in general. In a six-week pot experiment with rice plants, two Bangladeshi soils with contrasting SOC-to-reducible-Fe ($SOC:Fe_{ox}$) ratios were kept under a regime of alternate wetting and drying (AWD) or continuous flooding (CF), and were either amended with maize shoots or not. The $\delta^{13}C$ signatures of dissolved organic C and emitted $CH_4$ and $CO_2$ were used to infer the decomposition of added maize shoots ($\delta^{13}C = -13.0$ ‰) versus native SOC ($\delta^{13}C$

$= -25.4$ ‰ and $-22.7$ ‰). Addition of maize residues stimulated the reduction of Fe as well as the dissolution of native SOC, and the latter to a larger extent under CF, especially for the soil with the highest $SOC:Fe_{ox}$ ratio. Estimated Fe-bound SOC contents denote that stimulated SOC co-release during Fe reduction could explain this positive priming effect on SOC dissolution after the addition of maize. However, priming effects on SOC mineralisation to $CO_2$ and $CH_4$ were lower than for SOC dissolution, and were even negative under AWD for one soil. Enhanced reductive dissolution of Fe-bound SOC upon

exogenous OC addition therefore does not necessarily lead to stimulated SOC mineralisation. In addition, AWD irrigation was found to decrease abovementioned priming effects.





**Keywords**

Paddy soil, exogenous organic matter addition, anaerobic decomposition, priming effect, Fe-bound organic carbon, stable C isotope natural abundance

## 1 Introduction

Anaerobiosis in flooded paddy fields thoroughly affects soil chemical processes, as in absence of oxygen, the decomposition of organic carbon (OC) requires alternative terminal electron acceptors like manganese ($Mn^{4+}$), iron ($Fe^{3+}$), sulfate, acetate or carbon dioxide ($CO_2$). Anaerobic reduction of the latter two electron acceptors, which also depends on the production of dissolved organic carbon (DOC) as electron donor, leads to the production and emission of methane ($CH_4$) (Kögel-Knabner et al., 2010; Ponnamperuma, 1972). Obviously, irrigation management strongly affects the prevalence of aerobic versus anaerobic OC decomposition. The impact of irrigation management on soil processes is an extensively studied topic because of the demand to grow rice more efficiently with less water (Carrijo et al., 2017). Especially the adoption of alternate wetting and drying (AWD), a periodic drying and reflooding irrigation practice, is increasingly promoted (Lampayan et al., 2015). However, our understanding of how the alternation of redox conditions under AWD irrigation affects the decomposition of native SOC and exogenous OC in comparison with more consistent anaerobic conditions is still rather limited. This limited insight mainly stems from difficulties to discern native SOC and exogenous OC mineralisation based on soil gaseous emissions. In upland soils, assessments usually rely on $\delta^{13}C$ isotopic signatures of emitted $CO_2$ derived from C sources with distinct $\delta^{13}C$ (Hayes, 1983; Werth and Kuzyakov, 2010). However, in submerged soils, emission of $CH_4$ also needs to be accounted for. Since most $CH_4$ and $CO_2$ is emitted through rice plant aerenchyma transport, a realistic experiment necessitates the presence of growing rice plants, which further complicates the tracking of OC mineralisation in paddy soils. It is then important to account for C isotope fractionation during the OC decomposition process, mainly caused by the microbial community discriminating for or against $^{13}C$. In particular during anaerobic $CH_4$ production, the shift in $\delta^{13}C$ is typically large and moreover production pathway-dependent (Conrad, 2005; Sugimoto and Wada, 1993; Schweizer et al., 1999; Werth and Kuzyakov, 2010; Conrad et al., 2012). The $\delta^{13}C$ of emitted $CO_2$ is on the other hand influenced by $CO_2$ consumption during hydrogenotrophic $CH_4$ production (i.e. with $H_2/CO_2$ as substrate) and its partial storage as dissolved bicarbonate and $CO_2$. As a result, in absence of oxygen, it is much less straightforward to link the $\delta^{13}C$-$CO_2$ and $\delta^{13}C$-$CH_4$ of emissions to the $\delta^{13}C$ of the original C substrate (Conrad et al., 2012).

Furthermore, the incorporation of crop residues is a common practice in paddy soil (Ponnamperuma, 1984). It is known that exogenous OC addition can modify the rate of SOC decomposition in upland soils via so-called priming effects. Positive priming could for example result from soil microbial growth upon the addition of fresh OC as energy source, resulting in co-metabolism of biologically more recalcitrant SOC (Kuzyakov et al., 2000; Blagodatskaya and Kuzyakov, 2008). Positive as well as negative priming effects of OC addition on dissolution of native SOC and on its further mineralisation into $CO_2$ and





$CH_4$ has also been reported for flooded paddy soils (Bertora et al., 2018; Conrad et al., 2012; Ye et al., 2015; Ye and Horwath, 2017; Yuan et al., 2014), but again these processes are much less well understood. Indeed, considering the unique interlinkage

between anaerobic microbial activity and redox reactions with mineral soil oxidants, priming mechanisms could be quite different than in upland soils. For example, Bertora et al. (2018) and Ye and Horwath (2017) observed that rice straw addition enhances reductive dissolution of pedogenic Fe, and found indications that this in turn led to a stimulated co-release of OC originating from the Fe-OC complex. Fe hydroxides, which are often abundant in tropical soils, are indeed relevant sorbents of SOC owing to their high reactive specific surface area or through complexation (Wagai et al., 2013; Chen et al., 2020;

Kaiser and Guggenberger, 2000). Dissolution of OC from these minerals following the establishment of anaerobic conditions could thus be a considerable C-releasing process and may take place (i) by desorption due to a pH increase, caused by the consumption of protons during reduction reactions; or (ii) by co-release of sorbed OC along with reductive dissolution of short-range-ordered $Fe^{3+}$ minerals such as ferrihydrite and goethite (Grybos et al., 2009; Said-Pullicino et al., 2016). However, it is still unknown whether such stimulated native SOC dissolution would in turn significantly promote further conversion into $CO_2$

and $CH_4$. This would seem plausible as DOC is assumed to be one of the most bioavailable SOC fractions, especially under submerged conditions (Marschner and Kalbitz, 2003; Said-Pullicino et al., 2016). The indirect stimulation of SOC mineralisation by OC amendment through enhanced reduction of Fe in paddy soil may be expected to furthermore depend on the water regime, as this overridingly determines the extent of pedogenic Fe reduction and increase in pH. Lastly, opposite to co-dissolution of Fe and OC, DOC can likewise be adsorbed onto or can co-precipitate with the newly formed $Fe^{3+}$ oxides as

soon as the $E_h$ again increases and $Fe^{2+}$ is reoxidised (Sodano et al., 2017). Such C removal pathway might again strongly depend on irrigation management.

Our aim was to investigate how the addition of exogenous OC influences dissolution and mineralisation of native SOC in paddy soils in function of water management, with particular attention to the role of the co-release of Fe-bound SOC. To this end, patterns of DOC and emission of $CO_2$ and $CH_4$ were compared in a six-week pot experiment with living rice plants in a

tropical greenhouse. Two common irrigation practices were compared, i.e. the water-saving AWD practice next to continuous flooding (CF). We hypothesised that maize shoot addition would result in positive priming of native SOC mineralisation. This priming should be stronger under CF than under AWD, owing to enhanced pedogenic $Fe^{3+}$ reduction with stimulated net co-release of Fe-bound native SOC. Maize shoots were used as external OC source, because of the contrast of their $\delta^{13}C$ (C4 crop) with the $\delta^{13}C$ of the native SOC (largely C3-derived due to long-term rice cultivation) in both soils. A secondary aim was to

see if the proportion of SOC compared to pedogenic Fe, with the latter considered to be the main pool of readily reducible Fe, interacts with the effect of OC amendment on native SOC decomposition. Therefore, we compared two soils from Bangladeshi young floodplain paddy fields that were specifically chosen for their contrasting SOC to oxalate-extractable Fe ($Fe_{ox}$, as proxy for short-range-ordered Fe oxides that can be considered easily reducible (Postma, 1993)) ratio. We expected a stronger stimulation of native SOC release from the soil with the higher SOC:$Fe_{ox}$ ratio.





## 2 Material and Methods

### 2.1 Soils

Two young floodplain paddy soils (from soil series Balina and Sonatala) from northern Bangladesh were selected from a larger set of soils previously used by Akter et al. (2018) and Kader et al. (2013). The puddle layer (0 – 15 cm) was sampled at 15 locations per field in May 2014, and stored in air-dried, ground and sieved form. Both soils were selected based on their contrasting $SOC:Fe_{ox}$ ratios with other traits remaining largely similar (Table 1).

**Table 1.** Initial properties of the selected two paddy soil series and maize shoots used in the soil incubation experiment

| Parameter | Balina | Sonatala |
| --- | --- | --- |
| Soil texture[a] | Silty clay loam | Silt loam |
| Soil type[a] | Mollic Haplaquept | Aeric Haplaquept |
| Yearly cropping pattern[a] | Rice – fallow – fallow | Rice – fallow - rice |
| OC content (g kg$^{-1}$) | 14.1 | 22.4 |
| $\delta^{13}C$ (‰) | -25.37 | -22.70 |
| $Fe_{ox}$ (g kg$^{-1}$)[b] | 8.4 | 4.8 |
| $SOC:Fe_{ox}$ ratio (-) | 1.7 | 4.7 |
| pH-KCl[b] | 4.0 | 5.5 |
| pH-H$_2$O[c] | 6.1 | 6.9 |
| Total N (g kg$^{-1}$)[b] | 1.9 | 2.1 |
| $NH_4^+$-N (mg kg$^{-1}$)[d] | 8 | 3 |
| $NO_3^-$-N (mg kg$^{-1}$)[d] | 0 | 1 |
| C:N ratio | 7.4 | 10.7 |

[a] Data taken from Kader et al. (2013)

[b] Data taken from Akter et al. (2018)

[c] pH-H$_2$O by inserting a glass pH electrode in 1:5 soil–water extracts after 18 h
   equilibration

[d] Exchangeable mineral N determined in 1 M KCl extracts



X-ray diffraction analyses confirmed that both soils contained mica, vermiculite, chlorite and kaolinite, and Sonatala also some crystalline goethite (Kader et al., 2013). The presence of $Fe^{3+}$ in chlorite and vermiculite was confirmed by Mössbauer spectroscopy (Akter et al., 2018), but ferrihydrite and poorly crystalline goethite were the main $Fe^{3+}$ pools in both soils.

## 2.2 Pot experiment

A six-week pot experiment with living rice plants was run in the tropical greenhouse of the Faculty of Bioscience Engineering (Ghent University) in Melle, Belgium, from 23 April to 4 June 2018. The mean ambient temperature was 28 °C and the relative humidity was 82.2 %. PVC tubes (diameter: 18.5 cm | height: 25.0 cm) were filled with soil from either Balina (4.8 kg dry soil) or Sonatala (4.0 kg dry soil) until a height of 17.5 cm, matching their respective bulk densities, and were subjected to two contrasting irrigation treatments, as described below. Ground maize shoots were mixed into the soil at a dose of 4 g dry mass $kg^{-1}$ soil (i.e. 1.90 g C $kg^{-1}$) to three replicate pots per soil type and per irrigation regime. Maize shoots had an OC content of 474.4 g $kg^{-1}$, a $\delta^{13}C$ of -13.04 ‰ and a total N content of 13.0 g $kg^{-1}$. Two replicate pots were installed as controls without maize added, and two more replicates per irrigation and soil combination received maize (same dose of 4 g $kg^{-1}$) but had no living rice plants. Macronutrients were added once as basal fertilisation to all pots as urea, KCl and $Ca_3(PO_4)_2$, at doses of 60 kg N $ha^{-1}$, 40 kg K $ha^{-1}$ and 10 kg P $ha^{-1}$, respectively. Five 15-day-old rice seedlings were transplanted to each pot in a single hill. These seedlings were obtained by germinating rice seeds of BINA dhan14 (*Oryza sativa* L.), a Bangladeshi short-duration (maturity 120 - 130 days) dry season rice variety, and sowing those in seeding beds that were kept saturated in the tropical greenhouse for 15 days.

A standing water table of 2.5 cm was initially maintained in all pots by adding deionised water every one or two days to enable seedling establishment. Starting from 11 days after transplanting (DAT), the two different irrigation regimes were imposed. In pots under CF, the 2.5 cm water table was maintained until the end of the experiment. Pots under AWD, in contrast, were left to dry out until the water table dropped to between -8 cm and -14 cm, after which they were reflooded until a 2 cm standing water table was reached. Water table depths were monitored using a perforated tube (diameter: 3 cm) that was permanently installed in the pots. Soil drying was only due to evapotranspiration. Drying cycles took on average 6 ($\pm$ 3) days and shortened towards the end of the experiment.

## 2.3 Monitoring of biochemical soil parameters

The soil redox potential ($E_h$) was measured manually daily or every two days at 4 and 12 cm depth in eight pots with plants (i.e. 2 irrigation treatments x 2 soil series x with versus without maize). This was achieved by connecting a redox mV meter (Paleo Terra, Netherlands) to permanently installed platinum (Pt) probes and one Ag|AgCl reference electrode (Paleo Terra, Netherlands) per pot, and expressed relative to the standard hydrogen electrode. Soil pH was measured six times during the experiment in every pot at about 2 cm depth by inserting a glass pH electrode into the saturated soil.





To place $CH_4$ and $CO_2$ flux measurements in context, KCl-extractable ammonium ($NH_4^+$) and nitrate ($NO_3^-$) were determined at the start of the experiment (0 DAT), and at 11, 23, 30 and 45 DAT. For this, soil samples were taken by carefully inserting a tube (diameter: 1 cm) until 10 cm depth about four times at different locations near the side of each PVC pot. Soil samples

were then stored at -20 °C until analysis. The contents of $NH_4^+$ and $NO_3^-$ were determined with a continuous flow auto-analyser (Skalar, The Netherlands) after extracting soil samples with 1 M KCl at a 1:5 ratio.

At the end of the experiment, the above- and below-ground biomass of the rice plants was determined after drying at 60 °C for three days. The $δ^{13}C$ of SOC was determined before the experiment for solid native SOC and maize, and after the experiment in SOC, with an EA-IRMS (Automated Nitrogen Carbon Analyser – Solids and Liquids – coupled to a SerCon 20-20 IRMS,

Syscon Electronics, The Netherlands).

**2.4 Soil solution sampling and analyses of $Fe^{2+}$, $Mn^{2+}$, DOC and $δ^{13}C$-DOC**

To track dissolved Fe, Mn, Ca, Mg and DOC, soil solution samples (2 x 9 mL per pot) were taken ten times throughout the experiment. Solution samples were collected by connecting a pre-evacuated 12 mL glass vial through a needle, 2-way stopcock and extension tube to a porous MacroRhizon soil moisture sampler (Rhizosphere Research Products, The Netherlands), that

was permanently installed in each pot in a vertical position until about 9 cm depth. Solutions were analysed for their Fe, Mn, Ca and Mg concentrations by ICP-OES (Thermo Scientific, Unites States) without filtration, after acidifying the solutions with some drops of concentrated $HNO_3$ to redissolve precipitates. The sum of the molar concentrations of dissolved Fe and the increment in dissolved Ca and Mg (compared to their initial level) was taken as proxy for total reduced Fe since $Fe^{2+}$ dissolves easily, but readily exchanges with $Ca^{2+}$ and $Mg^{2+}$ on colloid surfaces (Saeki, 2004). In the same samples, the DOC

concentration and $δ^{13}C$-DOC were measured by FIA-IRMS (flow injection analysis with isotope ratio mass spectrometry, DELTA V Plus Advantage, Thermo Scientific, United States) after dilution in 0.85 % $H_3PO_4$, evacuation and ultrasonication to remove carbonates and dissolved gases. To account for small differences in soil to soil solution ratios, soil solution concentrations of Fe equivalents and DOC were for each time point converted to mmol dissolved Fe equivalents $kg^{-1}$ soil and mg C $kg^{-1}$ soil, respectively.

**2.5 Estimation of SOC bound to pedogenic Fe and its $δ^{13}C$**

We furthermore estimated the content and $δ^{13}C$ signature of SOC bound to weakly crystalline pedogenic Fe in the initial Balina and Sonatala soils based on release of SOC after reduction with hydroxylamine ($NH_2OH.HCl$), i.e. by assessing the change in OC content and $δ^{13}C$ of the solid soil samples with versus without $NH_2OH.HCl$ treatment. In all samples, particulate organic matter was firstly removed through ultrasonication at 60 J $ml^{-1}$ and subsequent wet sieving at 53 µm (Sleutel et al., 2007).

After drying the sieved suspensions, 0.8 g soil and 40 mL of a solution of 0.25 M $NH_2OH.HCl$ and 0.25 M HCl were mixed in a 85 mL Nalgene centrifuge tube, in analogy with Chao and Zhou (1983). The tubes were then kept in a water bath at 50 °C for 30 minutes with regular stirring. After centrifuging at 1000 g for 15 minutes, the supernatant was decanted, and traces of




NH2OH.HCl and HCl were removed by washing the residue soil three times with 40 mL demineralised water. Samples without the NH2OH.HCl treatment were treated identically, but the NH2OH.HCl solution was replaced by demineralised water. Finally, all samples were dried for four days at 40 °C, ground and analysed for their C content and $\delta^{13}$C by means of elemental analyser – isotope ratio mass spectrometry (EA-IRMS), i.e. an ANCA-SL (Automated Nitrogen Carbon Analyser – Solids and Liquids) interfaced with a 20-22 IRMS (Sercon Ltd., United Kingdom). The content of Fe-bound SOC was then estimated as the difference in SOC between treated samples and samples without NH2OH.HCl treatment. Analogous to Keeling (1958), the $\delta^{13}$C of Fe-bound SOC ($\delta^{13}C\text{-}SOC_{Fe}$) was calculated based on the following mass balance:

$$\delta^{13}C\text{-}SOC_{Fe} = \frac{\delta^{13}C\text{-}SOC_{untreated} * [C]\text{-}SOC_{untreated} - \delta^{13}C\text{-}SOC_{NH_2OH} * [C]\text{-}SOC_{NH_2OH}}{[C]\text{-}SOC_{untreated} - [C]\text{-}SOC_{NH_2OH}} \tag{1}$$

where $\delta^{13}C\text{-}SOC_{NH2OH}$ and $[C]\text{-}SOC_{NH2OH}$ refer respectively to the $\delta^{13}$C and C concentration of samples treated with NH2OH.HCl, and $\delta^{13}C\text{-}SOC_{untreated}$ and $[C]\text{-}SOC_{untreated}$ to those of samples without NH2OH.HCl treatment.

**2.6 Gaseous C fluxes**

In order to estimate decomposition of OC to gaseous CH4 and CO2, we measured emission fluxes and $\delta^{13}$C isotopic signatures of CO2 and CH4 at ten different moments for pots under CF (most frequently after onset of the experiment) and at twenty-six moments for pots under AWD (spread evenly over the course of the experiment). These extra measurements were required to sufficiently capture changing emissions during soil wetting and drying cycles. Gaseous efflux measurements were conducted by connecting a cavity ring-down spectrometer (G2201-*i* CRDS isotopic CO2/CH4 analyser, Picarro, United States) in a loop to an opaque PVC gas flux chamber (diameter: 18.5 cm | height: 25.0 cm) with internal battery-operated fan. For each measurement, this flux chamber was secured onto the pots by means of a PVC connection ring (diameter: 18.5 cm | height: 16.0 cm), and the headspace accumulation of CO2 and CH4 as well as the evolution of their $^{13}$C/$^{12}$C ratio (as opposed to the initial headspace air) was recorded every 4 seconds during 12 minutes. Starting from 28 DAT, an extra PVC extension ring (diameter: 18.5 cm | height: 25.0 cm) was added to the chambers because rice plants had outgrown the initial chamber.

CH4 and CO2 fluxes were determined as the slope of the accumulating headspace CH4 and CO2 concentrations in function of time, and were converted into a mass-based unit (mg C kg$^{-1}$ soil hour$^{-1}$) by means of the ideal gas law. Based on the conservation of mass, $^{13}$C/$^{12}$C ratios of the emitted CH4 and CO2 were determined as the intercept of the linear regression from the isotopic signature of the headspace gas in function of the reciprocal of the headspace concentration, also known as the Keeling plot method (Keeling, 1958). All $^{13}$C/$^{12}$C ratios were expressed relative to the international VPDB (Vienna Pee Dee Belemnite) standard as $\delta^{13}$C (‰). When rice seedlings were young and pots were flooded, a considerable part of the produced CO2 and CH4 appeared to be transported to the atmosphere through ebullition. In those cases, their accumulation in the headspace did not follow a linear course but was stepwise, with alternating time intervals of high and low accumulation rates, which were assumed to be dominated by ebullition versus by plant-mediated transport and molecular diffusion, respectively. When this





was the case, additional C emission fluxes and $\delta^{13}$C signatures were derived for shorter time intervals during which emissions were dominated by ebullition or by plant-mediated transport and diffusion.

Average daily fluxes were estimated from the measured fluxes by accounting for diurnal fluctuation of the actual temperature using a temperature dependency model:

Relative correction daily rate $= 1 - e^{\left((T_{avg}-T_{meas})*\ln(Q_{10})/10\right)}$ (2)

where $T_{avg}$ and $T_{meas}$ (in °C) respectively represent the average daily soil temperature and the temperature during measurement. The $Q_{10}$ temperature coefficient was set to 3.1 for $CH_4$ (Wang et al., 2015; Wei et al., 2021; Hattori et al., 2001) and to 2.3 for

$CO_2$ (Huang et al., 2015; Zhou et al., 2014a; Zhang et al., 2007; Wei et al., 2021). Overall, $T_{avg}$ was 29.5 ± 0.6 °C while $T_{meas}$ was 32.6 ± 1.0 °C, as gas efflux measurements were always carried out around noon.

**2.7 Source partitioning of gaseous and dissolved C**

The fractions of emitted $CO_2$ and $CH_4$ derived from added maize shoots (C4) ($f_{CO_2\,|\,maize}$ and $f_{CH_4\,|\,maize}$) versus from native SOC (C3) ($f_{CO_2\,|\,SOC} = 1 - f_{CO_2\,|\,maize}$ and $f_{CH_4\,|\,SOC} = 1 - f_{CH_4\,|\,maize}$) were inferred from $\delta^{13}$C signatures of emissions from (i)

the 'mixed pool' pots (to which maize was added); (ii) control pots without maize added ($\delta^{13}C$-$CO_2\,|\,SOC$ and $\delta^{13}C$-$CH_4\,|\,SOC$ as reference for the C3-derived emission endmember); and (iii) a shorter ancillary incubation with maize shoots as the only OC source ($\delta^{13}C$-$CO_2\,|\,maize$ and $\delta^{13}C$-$CH_4\,|\,maize$ as reference for the C4-derived emission endmember) (see *Supplementary material*). The latter two were used instead of the $\delta^{13}$C of respectively native SOC and maize-C in order to account for isotopic fractionation during microbial decomposition and emission of $CO_2$ and $CH_4$. Any effect of the presence of rice plants (plant

respiration or decomposition of plant photosynthates to $CO_2$ and $CH_4$) is accounted for by the fact that the control pots without maize added (i.e. $\delta^{13}C$-$CO_2\,|\,SOC$ and $\delta^{13}C$-$CH_4\,|\,SOC$) contained rice plants too. Based on the conservation of mass, the fraction of maize-derived $CO_2$ is for example (Werth and Kuzyakov, 2010; Rochette and Flanagan, 1997; Hayes, 1983):

$$f_{CO_2\,|\,maize} = \frac{\delta^{13}C\text{-}CO_2 - \delta^{13}C\text{-}CO_2\,|\,SOC}{\delta^{13}C\text{-}CO_2\,|\,maize - \delta^{13}C\text{-}CO_2\,|\,SOC}$$ (3)

The $CO_2$ or $CH_4$ fluxes derived from native SOC ($CO_2\,|\,SOC\text{-}derived$ or $CH_4\,|\,SOC\text{-}derived$) or maize ($CO_2\,|\,maize\text{-}derived$ or

$CH_4\,|\,maize\text{-}derived$) in the mixed pots were then obtained by multiplying emissions with $f_{CO_2\,|\,SOC}$ or $f_{CH_4\,|\,SOC}$ versus $f_{CO_2\,|\,maize}$ or $f_{CH_4\,|\,maize}$. Then, cumulative emissions of $CH_4$, $CO_2$, maize-derived C and native SOC-derived C were calculated.

We calculated a dimensionless priming effect (*PE*) coefficient of maize addition on SOC-derived $CO_2$ or $CH_4$ emission by comparing SOC-derived $CO_2$ and $CH_4$ fluxes with those from control pots without maize ($CO_2\,|\,SOC$ or $CH_4\,|\,SOC$) (Conrad et al., 2012). For example, relative priming of $CO_2$ emission was defined as:





$PE_{CO_2} = \frac{CO_{2 \,|\, SOC\text{-}derived}}{CO_{2 \,|\, SOC}} - 1$              (4)

In addition, the fraction of DOC derived from maize ($f_{DOC \,|\, maize}$) was calculated as well based on $\delta^{13}C$ signatures of DOC in amended pots ($\delta^{13}C\text{-}DOC$), of control pots without amendment ($\delta^{13}C\text{-}DOC_{SOC}$), of solid maize-C ($\delta^{13}C_{maize}$) and of solid SOC ($\delta^{13}C_{SOC}$), as follows:

$$f_{DOC \,|\, maize} = \frac{\delta^{13}C\text{-}DOC - \delta^{13}C\text{-}DOC_{SOC}}{\delta^{13}C_{maize} - \delta^{13}C_{SOC}}$$          (5)

DOC contents derived from SOC ($DOC_{SOC\text{-}derived}$) in amended soils were obtained by multiplying their total DOC contents with ($1 - f_{DOC \,|\, maize}$), and the relative priming effect of maize addition on SOC-derived C dissolution was then calculated in an analogous way to Eq. (4).

Because of the presence of rice plants, DOC contents as well as $CH_4$ and $CO_2$ emissions originate not only from native SOC versus maize shoots, but also from rice plant photosynthates. Having a similar $\delta^{13}C$ as SOC, rice plant-derived DOC, $CH_4$ and
$CO_2$ will contribute to what is partitioned as SOC-derived C, leading to overestimation of SOC decomposition. However, this contribution is believed to be very low in our experiment considering the young age of the rice plants. An estimate of the plant-derived DOC contents in our pots based on literature (Lu et al., 2004; Nguyen, 2003) was as low as 14 mg C kg$^{-1}$ soil, with therefore indeed low potential contributions to soil C emissions, in agreement with observations of Yuan et al. (2012) at 41 days, and comparison of our emissions from maize-amended pots with and without rice plants. SOC-derived emission
estimates based on isotopic mixing (Eq. 3) are hence further on assumed to be derived from native SOC decomposition.

### 2.8 Statistical analyses

Statistical tests were conducted with R 3.6.1 for Windows (R Core Team, 2019), with $P = .05$ taken as default significance level. Mostly, results for both soils (Balina and Sonatala) were analysed separately. The effect of irrigation treatment (starting from 11 DAT in case of time series) on measured variables was assessed by two-way or three-way ANOVA or by Kruskal-
Wallis tests with irrigation treatment and maize amendment as factors, and DAT as third factor in case of time series. In case of interaction between the irrigation management and maize amendment factors, the tests were repeated for data from maize-amended versus control pots separately. In case of interaction with DAT, differences between irrigation treatments were also assessed separately for individual points in time. Student's t-tests or Wilcoxon-Mann-Whitney tests were used to test the significance of priming effects for dissolution (i.e. comparing averages after integration over time from SOC-derived DOC in
maize-amended and unamended soil) as well as emission (i.e. comparing SOC-derived emissions from maize-amended and unamended soil), to compare if dissolution and emission priming effects differed from one another, and to test if the $\delta^{13}C$ of $CH_4$ and $CO_2$ differed when emissions were dominated versus not dominated by ebullition.





# 3 Results

## 3.1 Redox potential and iron reduction

Within one to two weeks the soil redox potential ($E_h$) generally decreased to ca. -250 to -150 mV (Fig. 1). The effect of the irrigation treatment on $E_h$ seemed to depend on both soil and addition of maize, but temporary increases in $E_h$ were observed upon soil drying. The $E_h$ was more responsive to wetting and drying at 4 cm than at 12 cm depth. Maize addition did not particularly induce a lower $E_h$, irrespective of soil type, though it did stimulate reductive dissolution of Fe ($P < .001$ for both soils between 2 and 11 DAT) during the observed initial increase in dissolved Fe equivalents (Fig. 2). Similar trends were

observed for $Mn^{2+}$ (data not shown). When maize was added, Fe concentrations after 11 DAT dropped to lower levels in case of AWD compared to CF ($P < .001$ for Balina and $P = .04$ for Sonatala). In control pots without maize shoots added, there was no significant effect of irrigation treatment on the decrease in dissolved Fe.





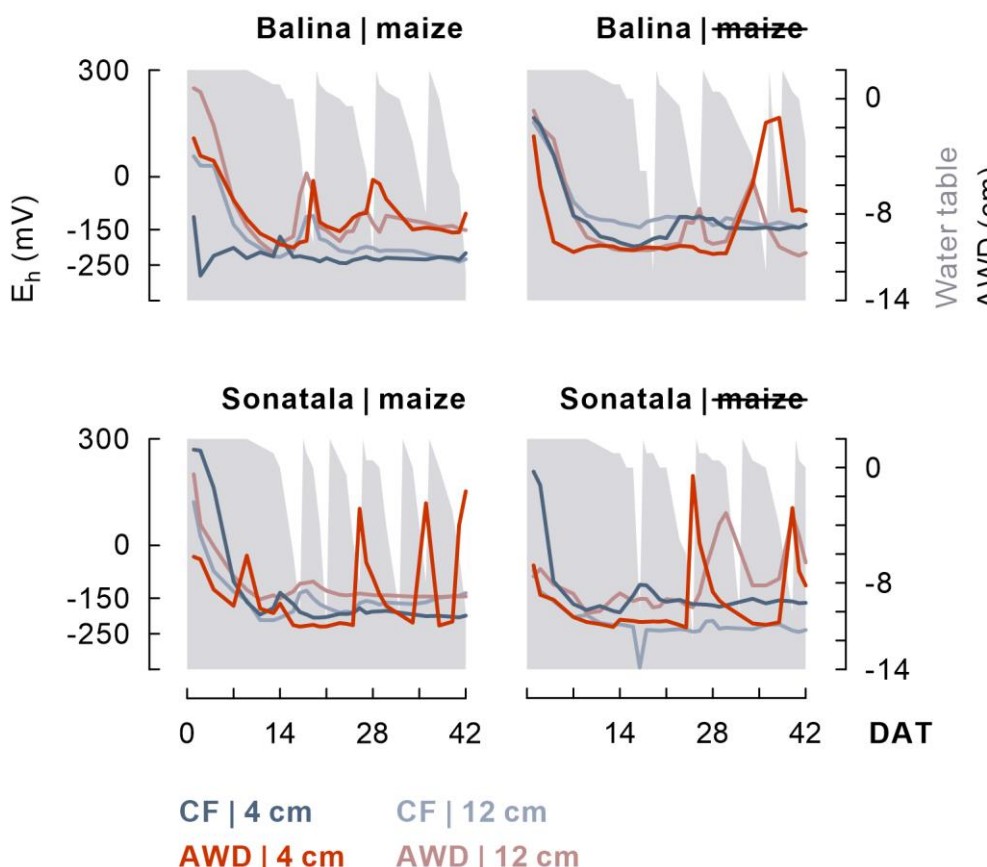

**Figure 1.** After a general initial drop, the soil redox potential ($E_h$) (values per replicate at 4 cm and 12 cm depth) depended to some extent on the irrigation treatment in both paddy soils (Balina and Sonatala). Water table fluctuations in case of AWD are indicated in grey.





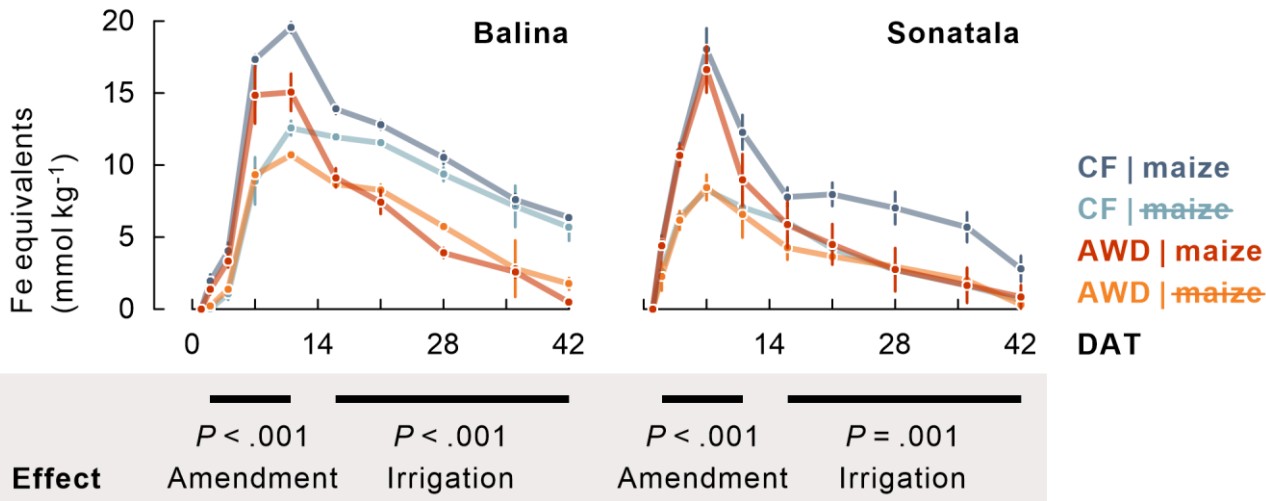

**Figure 2.** Maize addition stimulated reductive dissolution of Fe (average ± standard error of three or two replicates, calculated as sum of dissolved Fe and increments in dissolved Ca and Mg), and $Fe^{2+}$ disappeared to a larger extent from the soil solution under AWD than under CF in maize-amended pots. The p-values indicate the effect of maize addition on the time series between 2 and 11 DAT, and the effect of irrigation treatment in maize-amended pots on the time series between 16 and 42 DAT.

### 3.2 Exchangeable $NH_4^+$, pH and rice plant biomass

Concentrations of exchangeable $NH_4^+$ increased until 11 DAT, partly owing to the initial urea additions of 39 mg N kg$^{-1}$ and 47 mg N kg$^{-1}$ to the Balina and Sonatala soils, respectively. Thereafter they remained more or less constant (Supplementary Fig. S2). Overall, exchangeable $NH_4^+$ levels reached in the Balina soil were about double of those for Sonatala. Effects of the presence of rice plants or irrigation treatment were at times significant but tended to be small and are not further discussed in detail. $NH_4^+$ levels were most of the times significantly higher with maize added than without, probably because of net N mineralisation. Concentrations of $NO_3^-$ were always minor, i.e. below 1 mg N kg$^{-1}$ (data not shown).

The soil solution pH of the Balina soil increased from an initial value of 6.1 to 6.8 (± 0.1) after 17 DAT and then remained constant onwards (Supplementary Fig. S3). In the Sonatala soil, pH remained nearly constant throughout (initially 6.9, then 6.8 ± 0.1 starting from 17 DAT).

After 42 days, rice plants had reached a height of 76 ± 5 cm, with an above-ground dry biomass of 2.7 ± 0.7 g per pot and a below-ground dry biomass of 0.6 ± 0.3 g per pot. There was no effect of soil type or irrigation practice on the above-ground biomass. Below-ground biomass was neither affected by soil type but was significantly higher for plants grown under CF (0.8 ± 0.3 g per pot) than under AWD (0.4 ± 0.2 g per pot) ($P < .001$).





### 3.3 Organic carbon dissolution

Extremely high total DOC peaks were reached 1 to 2 weeks after submergence in all treatments (maximally 1912 and 1329 mg C kg⁻¹ soil for maize-amended Balina and Sonatala soils, respectively), which were also much higher with maize added ($P$ < .001 for time series until 11 DAT). Not only maize-derived DOC (data not shown), but also enhanced dissolution of native SOC caused these DOC peaks (Fig. 3). Over the course of the experiment, SOC-derived DOC was indeed higher with maize added than without in both soils, which was significant for Sonatala under CF (averaged concentrations over the course of the season: $P$ = .003). In other words, there was a positive priming effect on SOC dissolution (i.e. stimulation) during most of the incubation. This priming effect was irrespective of the irrigation regime in the Balina soils ($P$ = .97), while it was stronger under CF than under AWD in the Sonatala soils ($P$ < .001) starting from 16 DAT. Priming of SOC dissolution was also stronger for Sonatala than for Balina ($P$ < .001). Dissolution of maize-C was in contrast nearly identical in both soils, with an average peak concentration of 811 mg C kg⁻¹ at 7 DAT, followed by a near-complete disappearance of maize-derived DOC around 16 DAT (< 79 mg C kg⁻¹) (data not shown).







**Figure 3.** Dissolution of native SOC (average ± standard error of three or two replicates) was stimulated upon addition of maize, as it appears when comparing with DOC in unamended pots (indicated in brown). P-values of the priming effect significance using average values per season (in the figure) and of the irrigation treatment effect on the priming effect time series starting from 16 DAT (below the figure) are indicated.

**3.4 Gaseous carbon emission**

Gaseous C emissions consisted on average of 31 % $CH_4$ and 69 % $CO_2$. Irrigation management did not affect the cumulative C emissions ($CO_2 + CH_4$), neither $CO_2$ nor $CH_4$ emissions individually in case of the Balina soil (Fig. 4). For the Sonatala soil, however, the effect of irrigation depended on whether or not the soils were amended with maize. With maize added, total gaseous C ($P = .04$) and $CH_4$ ($P = .03$) emissions were higher under CF than AWD, but $CO_2$ emissions were not significantly affected. On the other hand, in unamended Sonatala pots, total C ($P = .02$) and $CO_2$ ($P = .003$) emissions were higher under





AWD than under CF, while $CH_4$ emission was not significantly different. $CH_4$ and $CO_2$ emissions were in general higher with

maize added than without ($P < .001$). Total gaseous C emission was furthermore lower from the Sonatala than the Balina soil

305    ($P = .01$). In general, gaseous C emissions were similar with or without living rice plants in pots with maize added ($P = .23$).

Both the addition of maize ($P < .001$) and the implementation of CF ($P < .001$) led to more gaseous C emission via ebullition:

$60 \pm 15$ % of C was emitted through ebullition under CF compared to only $37 \pm 11$ % under AWD. With maize added, $56 \pm$

16 % of C was emitted via ebullition, compared to only $37 \pm 12$ % for unamended control pots.

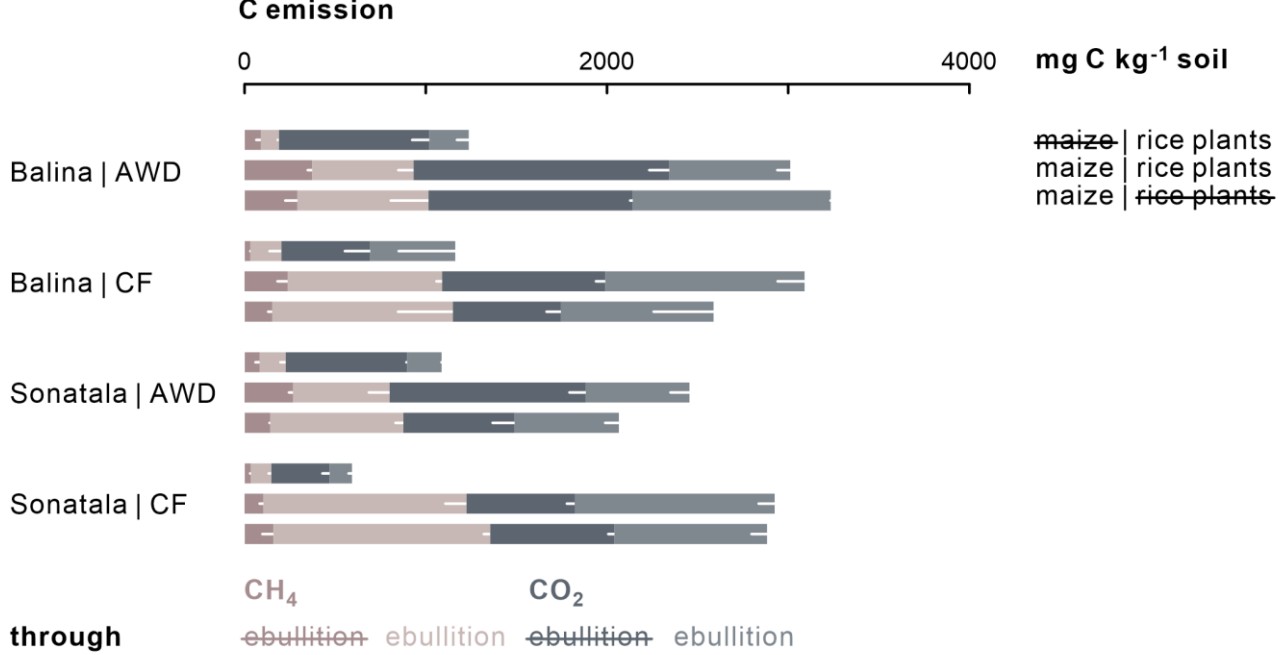

310    **Figure 4.** Total cumulative C emissions (average - standard error of three or two replicates, represented by white bars) from maize-amended

soils were two to fourfold of those from unamended pots ($P < .001$). In addition, relatively more C was emitted through ebullition under CF

($P < .001$) and upon addition of maize shoots ($P < .001$).

In addition, the course of $CH_4$ and $CO_2$ fluxes seemed linked to the wetting and drying cycles under AWD (Fig. 5) as (i) more

gaseous C was emitted when the soils dried; (ii) most ebullition transport occurred when the water table was at the soil surface,

315    usually a few days before rewetting; and (iii) $\delta^{13}$C-$CO_2$ signatures increased as water tables declined.



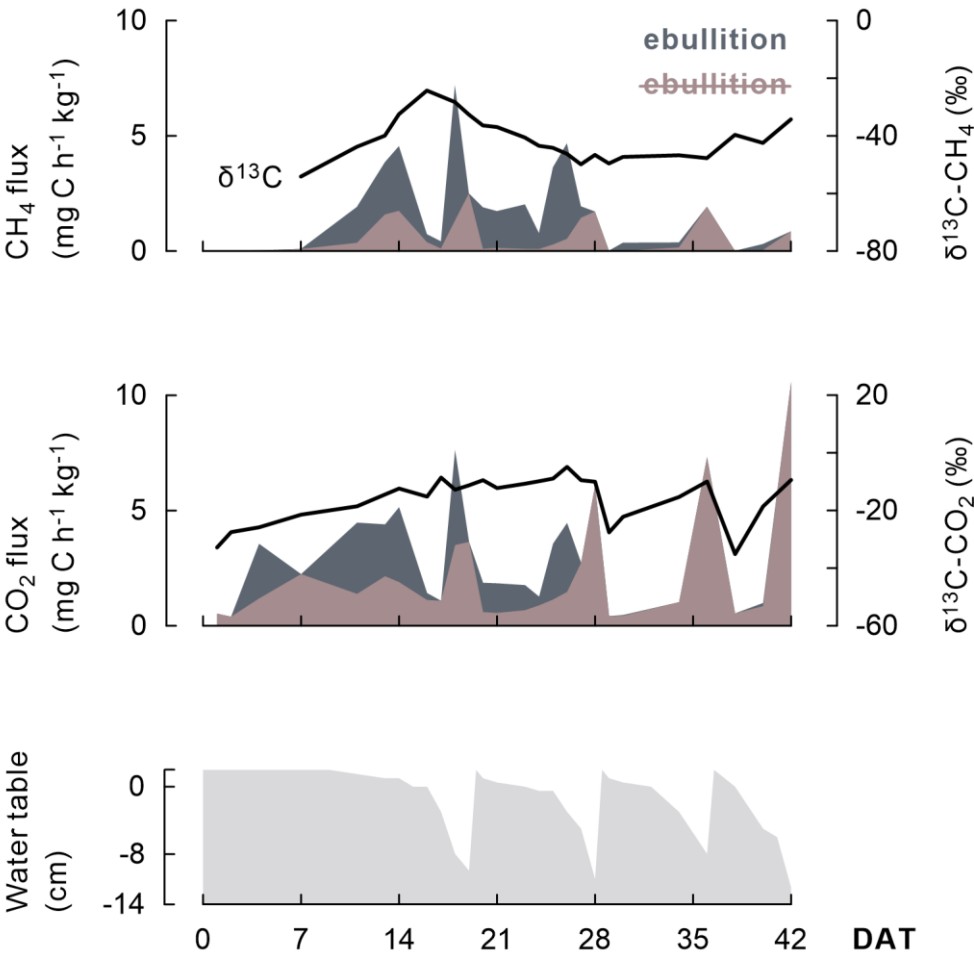

**Figure 5.** As this representative example (one replicate of the Balina soil with maize shoots added and under AWD) shows, the progression of $CH_4$ and $CO_2$ emission fluxes (shaded) and even their $\delta^{13}C$ signature (black line) appear linked to the change in water table depth under AWD. A distinction is made between emission fluxes dominated by ebullition (dark grey shaded) versus by diffusion or plant transport (brown shaded). Transport through ebullition was particularly considerable when rice plants were juvenile.

CH$_4$ and CO$_2$ emissions were clearly more enriched in $^{13}C$ when maize was added (Fig. 6 and Supplementary Fig. S1), evidently indicating the contribution of maize C decomposition. The $\delta^{13}C$ of CH$_4$ increased to a distinct maximum at 16 DAT (Balina) or 13 DAT (Sonatala) (Supplementary Fig. S1). In control pots without maize, the $\delta^{13}C$ of emitted CH$_4$ was higher for Sonatala than for Balina, in line with the difference in $\delta^{13}C$ of native SOC (Fig. 6). Implementation of AWD to control pots resulted in higher $\delta^{13}C$ signatures of emitted C (CO$_2$ + CH$_4$) compared to CF ($P = .02$). Isotopic fractionation took place during mineralisation of maize and native SOC into CH$_4$ and CO$_2$ (Fig. 6 and Supplementary Fig. S1). Lastly, the $\delta^{13}C$ of CH$_4$ ($P =$





.02) and $CO_2$ ($P < .001$) was on average 2.62 ‰ and 3.45 ‰ higher when emissions were dominated by ebullition than when they were dominated by plant transport and diffusion (data not shown).

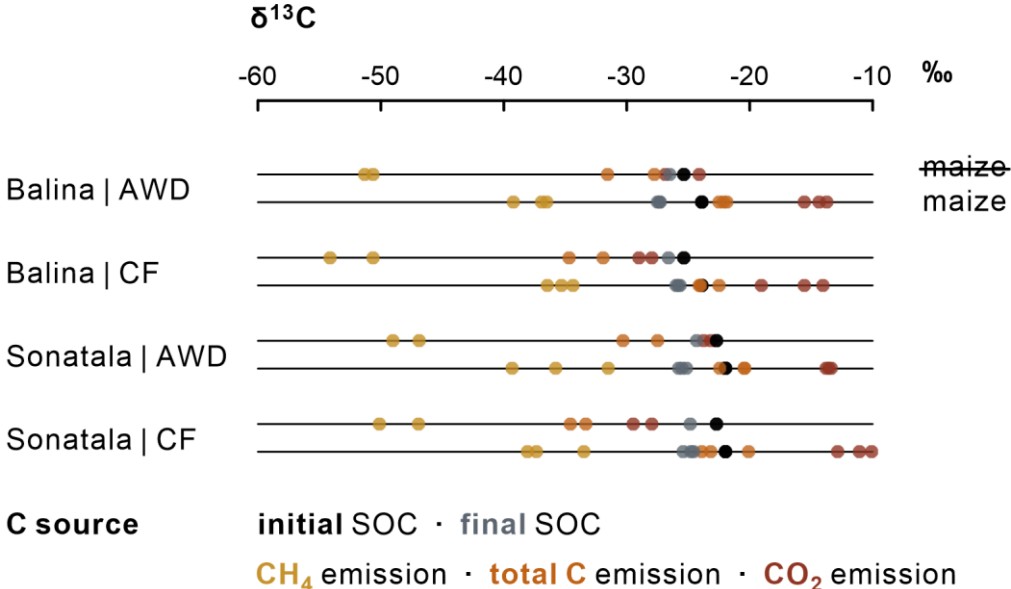

**Figure 6.** In absence of maize, the overall $\delta^{13}C$ signature of C emissions (values per replicate) was lower (i.e. less $^{13}C$-enriched) under CF than under AWD ($P = .02$).

### 3.5 Source partitioning of emitted $CO_2$ and $CH_4$

The fractions of maize-derived and SOC-derived $CO_2$ and $CH_4$ were calculated based on isotopic mixing, considering the $\delta^{13}C$ of SOC and maize, and accounting for isotopic fractionation during decomposition to $CO_2$ and $CH_4$ (Supplementary Fig. S1). Surprisingly, the $\delta^{13}C$-$CH_4$ of emissions from an ancillary C4-reference incubation with only maize and no native SOC as OC ($\delta^{13}C$-$CH_{4\,|\,maize}$) was usually lower (instead of higher) than from control pots with no maize added ($\delta^{13}C$-$CH_{4\,|\,SOC}$) (Supplementary Fig. S1). The difference in isotopic signature between maize and SOC was therefore not further reflected in a likewise difference in $\delta^{13}C$ of the $CH_4$ derived from both sources. In addition, the $\delta^{13}C$-$CH_4$ was often higher than the $\delta^{13}C$ of both endmembers in case of the maize-amended Balina and Sonatala soils. For these reasons, the fraction of $CH_4$ emission derived from maize ($f_{CH_4\,|\,maize}$) could unfortunately not be reliably deduced, in spite of the high measurement frequency across the experiment. As a best estimate, we therefore assumed that the contributions of emitted $CH_4$ from maize decomposition would equal $f_{CO_2\,|\,maize}$. After about 16 DAT, this derived fraction of maize-derived $CO_2$ increased to about 100 % for most of the pots (data not shown). The calculated cumulative maize-derived $CO_2$ and $CH_4$ emissions ranged between 1737 and 2757 mg C $kg^{-1}$, in most cases exceeding the theoretical maximum of 1898 mg C $kg^{-1}$, i.e. the amount of initially added maize.




Therefore, in those cases we instead consider complete degradation of all the initially added maize for further calculation of cumulative SOC-derived C emissions, which seems reasonable considering the complete disappearance of dissolved maize-C around 16 DAT (Fig. 3), and the fact that $\delta^{13}C$ of SOC was not higher after than before the experiment (Fig. 6).

The effect of maize addition on SOC-derived C emission was negligible in case of Balina, while for Sonatala, it depended on the irrigation treatment (Fig. 7). CF irrigation led to a positive priming ($P = .05$) (i.e. stimulated SOC-derived gaseous C

emissions upon maize addition), whereas negative priming on SOC-derived C emissions in maize-amended soil occurred under AWD ($P = .01$). In addition, priming on SOC-derived C emission was higher under CF than AWD for Sonatala ($P = .004$).

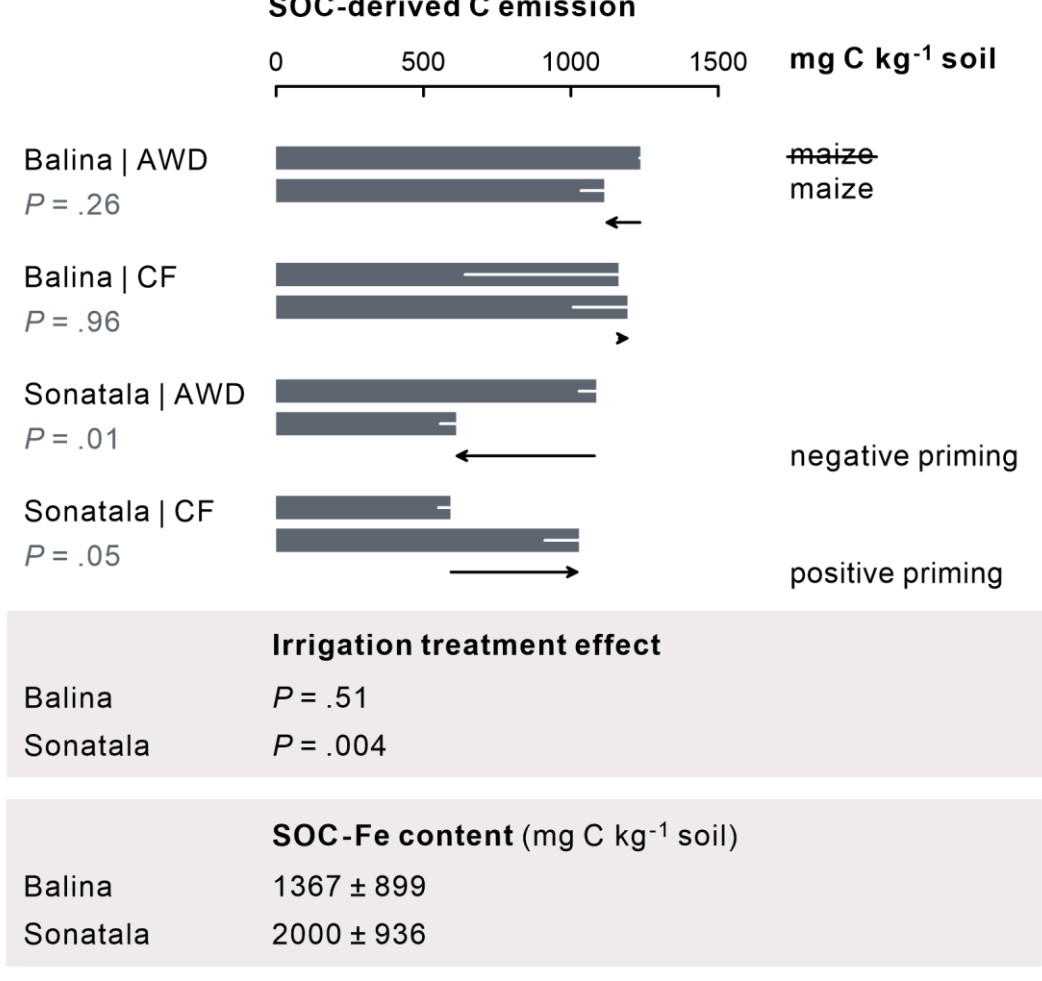

**Figure 7.** For the Sonatala soil, maize addition led to positive priming of gaseous C ($CO_2 + CH_4$) emission under CF and negative priming under AWD. P-values of the priming effect significance (in the figure) and of the irrigation treatment effect on the priming effect (below the

figure) are indicated. Standard errors (three or two replicates) are indicated. Fe-bound SOC levels (average ± standard error, estimated by



means of $NH_2OH.HCl$) exceed the positive priming effect of Sonatala under CF, so stimulated Fe-C dissolution and emission has the potential to explain this priming.

## 4 Discussion

### 4.1 Are Fe reduction and SOC dissolution linked?

With the onset of anaerobic conditions, exemplified by a decreasing soil $E_h$ (Fig. 1) there was a strong increase in dissolved Fe (Fig. 2), which shows that part of the soil Fe was quickly reduced and dissolved. Alongside, DOC concentrations increased rapidly until maxima at 11 DAT (Balina) and at 4 – 7 DAT (Sonatala). The synchrony of OC dissolution and Fe reduction patterns prompts to question if both processes are physically linked. Said-Pullicino et al. (2016) suggested that observed DOC increases could have been due to the co-release of OC during reductive dissolution of Fe oxyhydroxides. With an adapted

$NH_2OH.HCl$ extraction procedure, we estimated that the amount of C bound onto reducible $Fe^{3+}$ minerals was 1367 mg kg$^{-1}$ for Balina and 2000 mg kg$^{-1}$ for Sonatala (Fig. 7), constituting approximately 9 % of SOC. Release of this substantial Fe-bound SOC pool might have been a crucial source of DOC, but other important C dissolution pathways need to be considered as well. Firstly, a rise in pH upon soil reduction under anaerobic conditions can lead to SOC desorption (Grybos et al., 2009) but this mechanism was most probably unimportant here because of the initially high pH that did not considerably change during the

incubation, especially in the Sonatala soil (Supplementary Fig. S3). Secondly, rhizodeposition from rice plants can also cause an increase in DOC (Said-Pullicino et al., 2016; Lu et al., 2004). However, based on values from Lu et al. (2004) and Nguyen (2003), maximum DOC contents derived from rice plant photosynthates in our soils are estimated to be around 14 mg C kg$^{-1}$ soil, which is again limited since rice plants were juvenile during the experiment. Lastly, fermentation of SOC and added maize into water-soluble organic metabolites must obviously have largely contributed to the accumulation of DOC. As such,

based on DOC patterns alone it is not possible to unequivocally state that reductive dissolution of Fe with co-release of Fe-bound OC drove DOC build-up.

    The stable isotope approach then allowed to discriminate between SOC-derived and maize-derived DOC. Maize addition roughly doubled dissolution of native SOC (Fig. 3). Since also Fe reduction was stimulated upon the addition of maize (Fig. 2), it seems likely that, next to stimulated SOC fermentation, indeed promoted SOC co-release upon enhanced Fe reduction

was a potentially important priming mechanism, in agreement with the observations of Ye and Horwath (2017) and Bertora et al. (2018). Dissolution of all Fe-bound SOC present might produce a DOC peak of approximately 584 mg C kg$^{-1}$ (Balina) or 854 mg C kg$^{-1}$ (Sonatala), when roughly comparing it to the DOC production and DOC mineralisation kinetics of maize (where we know that 1898 mg added maize-C kg$^{-1}$ yielded a peak of 811 mg C kg$^{-1}$ maize-derived DOC). While just a tentative approximation, it becomes clear that the magnitude of SOC-derived DOC peaks suffices to explain the observed increment in

SOC-derived DOC with maize added (Fig. 3).





Comparison of the $\delta^{13}C$ of Fe-bound SOC and DOC could further contribute to the quantification of DOC derived from pedogenic Fe. Attempts to measure $\delta^{13}C$ of $NH_2OH.HCl$ extracts directly by means of FIA-IRMS unfortunately failed owing to interference of the matrix, even after trying to remove $NH_2OH.HCl$ and $Cl^-$ by oxidation and precipitation. However, from the difference in soil $\delta^{13}C$ before and after $NH_2OH.HCl$ extraction, the $\delta^{13}C$ of Fe-bound SOC could be estimated indirectly

with reasonable uncertainty, i.e. -27.8 ± 0.7 ‰ for Balina and -23.3 ± 3.3 ‰ for Sonatala. Apparently, Fe-C was a bit more depleted in $^{13}C$ than bulk SOC (Table 1). In control pots, with only Fe-bound SOC and bulk SOC as C sources, $\delta^{13}C$ signatures of DOC were also lower, supporting the assumption that DOC partly consists of released Fe-bound C. To robustly derive the contribution of potentially enhanced Fe-bound C release to stimulated SOC dissolution in maize-amended pots, three-source partitioning with an additional contrast in $^{14}C$ next to the existing variation in $^{13}C/^{12}C$ would be required (when rice plants can

be disregarded).

### 4.2 Extent of priming of SOC mineralisation as compared to SOC dissolution

Our objective was then to more specifically assess whether the stimulated SOC dissolution would be reflected in enhanced native SOC-derived mineralisation to $CO_2$ and $CH_4$. As it turns out, native SOC-derived gaseous C (i.e. $CO_2 + CH_4$) emissions were not significantly modified by addition of maize in the Balina soil (Fig. 7). For Sonatala, there was however positive

priming (i.e. more SOC-derived C emissions in soil with maize addition than in soil without) under CF irrigation ($P = .05$) and a negative priming under AWD ($P = .01$). For the CF treatment, the discussed stimulation of native SOC dissolution, probably partially by enhanced co-release of Fe-bound SOC, may have promoted SOC mineralisation upon maize addition. Yet, priming effect coefficients were overall larger for dissolution than for gaseous C emission in Sonatala ($P < .001$) and not significantly for Balina ($P = .18$) (Fig. 8), which suggests that the enhanced SOC dissolution would not have resulted in a proportionate

SOC mineralisation. Hanke et al. (2013) likewise found that extra solubilised C after paddy soil submergence did not cause a proportional rise in gaseous C emissions. They assumed that the availability of alternative terminal electron acceptors was limiting mineralisation. However, as SOC mineralisation (indicated by SOC-derived $CO_2$ and $CH_4$ emission) was actually slower under AWD than under CF in the Sonatala soil, electron acceptor availability must not really have limited priming of SOC-derived C emission. It instead seems more likely that preferences of microorganisms that decompose DOC explain the

observed smaller priming of soil C emission compared to dissolution. DOC-decomposing microorganisms must have preferred the abundantly available maize-derived DOC at the expense of SOC-derived DOC mineralisation in maize-amended soils. This hypothesis is supported by the faster removal of maize-derived compared to SOC-derived DOC (data not shown). Maize shoots and native SOC obviously differ in their biochemical quality, where SOC in paddy soils is generally quite aromatic with an accumulation of hardly degradable lignin components (Olk et al., 2002; Zhou et al., 2014b). In fact, phenol

accumulation in paddy soil in part explains their larger topsoil OC content compared to upland counterparts (Chen et al., 2020). The energy yield of oxidation of such reduced SOC compounds (phenols and lipids) becomes low when coupled to Fe reduction





(Keiluweit et al., 2017), and therefore it is logical that maize-derived DOC would have been preferred over SOC-derived DOC
as microbial substrate.

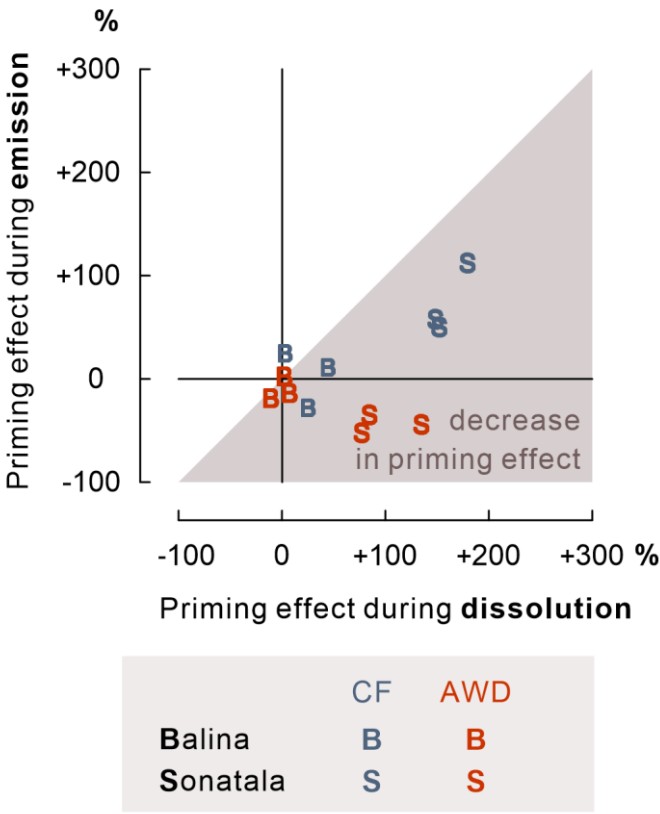

**Figure 8.** For Sonatala ($P < .001$) and not significantly for Balina ($P = .18$), the priming effect on gaseous C ($CO_2 + CH_4$) emission was lower than on SOC dissolution.

### 4.3 Influence of irrigation management and soil type

Furthermore, we wanted to verify whether irrigation management plays a role in the priming effect on SOC-derived dissolution
and mineralisation. It is of interest that the enhancement of SOC dissolution by maize addition was stronger under CF than
AWD, especially for the Sonatala soil (Fig. 3). Under permanently flooded conditions, the addition of maize was probably
more effective in promoting anaerobic microbial fermentation resulting in dissolution of SOC, as the biomass of anaerobes
could grow under less disturbed conditions than in case of a regime with repeated soil drying. This irrigation treatment effect
on DOC priming was also manifested in higher priming on SOC emission under CF than AWD in case of Sonatala ($P = .004$)
(Fig. 7). Furthermore, the decrease in priming effect was slightly more pronounced for AWD than CF for Sonatala ($P = .08$)
(Fig. 8). The negative priming of gaseous C emission under AWD in Sonatala was therefore possibly not only due to a lesser





stimulation of SOC-derived dissolution under AWD than under CF. On top of that, the microbial community under AWD might have been less adapted to anaerobic mineralisation than under CF, resulting in an even larger preference for maize-derived than SOC-derived DOC.

Furthermore, the effect of irrigation management on overall gaseous emissions was rather limited, especially for Balina soil, and surprisingly partially determined by ebullition. With no maize added, $CO_2$ emission ($P = .003$) was lower under CF than under AWD for Sonatala (Fig. 4). Indeed, SOC mineralisation is often retarded under anaerobic conditions (Kögel-Knabner et al., 2010; Sahrawat, 2004). However, also $CH_4$ effluxes were surprisingly lower under CF, while the more reducing conditions compared to under AWD were expected to stimulate $CH_4$ production (Jiang et al., 2019; Peyron et al., 2016). One cause for this contrasting outcome may be the often considerable temporal storage of produced $CH_4$ and $CO_2$ in submerged CF pots. Indeed, transport of $CH_4$ and $CO_2$ in paddy soils usually proceeds predominantly via rice plant aerenchyma, whereas diffusion-based outgassing is retarded by permanently standing water. With rice plants still juvenile and relatively limited $CO_2$ and $CH_4$ production in the unamended pots, it is thus possible that a larger part of these gases was stored dissolved or as gas bubbles in the CF pots. With AWD on the other hand, the capacity of the standing water to store gases decreases as the water table regularly goes down (Green, 2013; Tokida et al., 2013). Transport of the produced $CH_4$ and $CO_2$ to the atmosphere was indeed clearly facilitated by episodic flushes, e.g. through ebullition, when the water table decreased (Fig. 5), in line with the observations of Yagi et al. (1996) and Han et al. (2005). With maize added, on the other hand, $CH_4$ emission ($P = .03$) was 53 % higher under CF than AWD in case of the Sonatala soil. The much faster production of $CH_4$ compared to in unamended pots likely led to oversaturation of the dissolved $CH_4$ pool, resulting in less restricted $CH_4$ emission. For Balina, on the contrary, the irrigation regime did not significantly affect $CO_2$ and $CH_4$ emission. This all in all low effect of water management on gaseous C emissions in the Balina soil was in line with its limited impact on the progression of $E_h$ at 12 cm depth (Fig. 1), pH (Supplementary Fig. S3) and the fact that exchangeable $NH_4^+$ levels (Supplementary Fig. S2) only differed during the final two weeks.

Lastly, the contrast in Fe-bound SOC estimates between Sonatala and Balina corresponded with the difference in their respective SOC:$Fe_{ox}$ ratios (4.7 versus 1.7), and moreover with the higher SOC-derived DOC levels and stronger priming of SOC dissolution in Sonatala compared to Balina. Ye et al. (2016) found that a soil with less electron acceptors gave rise to stronger priming on $CH_4$ emission in a laboratory incubation under submerged conditions, which corresponds with our findings on C ($CH_4$ + $CO_2$) emission priming for the CF treatment. However, the higher SOC:$Fe_{ox}$ ratio cannot immediately explain the more pronounced decrease in priming effect on C emission compared to dissolution for Sonatala than for Balina (P < .001), except if the DOC derived from pedogenic Fe would be more recalcitrant, leading to a bigger preference of DOC decomposers for maize-derived DOC. Soil type also influenced temporal evolutions of biochemical parameters, but only to some extent. For Sonatala, redox-related processes took place three to five days earlier than for Balina, such as: (i) the initial decline of $E_h$ (Fig. 1); (ii) the peak in dissolved Fe (Fig. 2); (iii) the peak in DOC, in particular in SOC-derived DOC (Fig. 3); and (iv) the "peak" in emission $\delta^{13}C$-$CH_4$ signatures (Supplementary Fig. S1). This probably all relates to the higher electron surplus (i.e.



higher amount of electron donor as compared to the main electron acceptor $Fe^{3+}$) in combination with a lower $Fe_{ox}$ content for

Sonatala (with its higher $SOC:Fe_{ox}$ ratio), resulting in a faster decrease in soil $E_h$. In Sonatala, it is likely that microbes quickly needed to couple OM oxidation with reduction of alternative electron acceptors other than $Fe^{3+}$, so that $E_h$ dropped below the level enabling $CH_4$ production (-150 mV) after 4 days on average for Sonatala while only after 9 days for Balina. In general, the impact of irrigation management and maize residue addition on DOC release and emissions seemed to depend on the soil type (in particular the $SOC:Fe_{ox}$ ratio), and soil type therefore seems to be an equally important factor controlling the

progression of reductive processes.

### 4.4 Using an isotope mass balance to infer native SOC mineralisation in paddy soils

Because of different types of isotopic fractionation, most importantly caused by microbial discrimination for or against $^{13}C$ as substrate, source partitioning of gaseous emissions was not straightforward. Unlike the results of Conrad et al. (2012) but in agreement with the observations of Ye et al. (2016), isotopic fractionation was different for maize than for SOC. For example,

we observed a very consistent initial enrichment in $^{13}C$-$CH_4$ over time until strikingly high $\delta^{13}C$-$CH_4$ signatures of -22.4 ± 2.9 ‰ (Balina | 16 DAT) and -26.4 ± 4.4 ‰ (Sonatala | 13 DAT), with thereafter $^{13}C$ depletion (Supplementary Fig. S1 and Fig. 5). As such trends were inexistent for $\delta^{13}C$-$CO_2$, these $\delta^{13}C$ evolutions are likely the result of gradually changing dominant $CH_4$ production pathways. Indeed, $CH_4$ can be produced with acetate, $CO_2$ or other methylated compounds as precursors. Among any of these precursors, the lowest fractionation (i.e. resulting in $CH_4$ with the least negative $\delta^{13}C$) occurs with the

acetate-dependent pathway, resulting in a $\delta^{13}C$-$CH_4$ as high as -27 ‰ (relative to VPDB) (Conrad, 2005). The even further enrichment of $^{13}C$-$CH_4$ in the maize-amended plots can be explained by the higher $\delta^{13}C$ of maize (Fig. 6). Another explanation for constantly changing $\delta^{13}C$-$CH_4$ could be that the contributions of various gas transport mechanisms and associated isotopic fractionation would have changed over time. The $\delta^{13}C$-$CO_2$ increased as water tables declined during AWD cycles, particularly towards the end of the experiment (Fig. 5). In the unamended pots, the $\delta^{13}C$ of $CO_2$ (P = .02) was also higher under AWD than

under CF (Fig. 6). Both observations clearly suggest that change in transport fractionation affected the evolving $\delta^{13}C$-$CO_2$ and $\delta^{13}C$-$CH_4$ patterns. Interestingly, continuous CRDS-based recording of $CO_2$, $CH_4$ and their $\delta^{13}C$ allowed to discern sudden ebullition events over the course of a measurement. Emissions dominated by ebullition indeed displayed higher $\delta^{13}C$-$CH_4$ but also higher $\delta^{13}C$-$CO_2$ signatures than emissions during which ebullition appeared absent. The evolving contribution of diffusion and plant-mediated versus ebullition transport throughout the experiment thus must have altered the $\delta^{13}C$ of emitted

$CO_2$ and $CH_4$. In case of $CH_4$, it is indeed well known that $\delta^{13}C$ isotopic fractionation not only takes place during plant-mediated transport (Zhang et al., 2014; Zhang et al., 2015) but also during gas bubble formation (Zhang et al., 2014). Our results suggest that also $\delta^{13}C$ of emitted $CO_2$ is impacted by the contribution of ebullition transport. Using pots without maize added, we quantified the overall isotopic fractionation between SOC and SOC-derived $CO_2$ and $CH_4$. We then also attempted to quantify isotopic fractionation between maize-C and maize-derived $CO_2$ and $CH_4$ by means of an ancillary incubation

experiment (Supplementary Fig. S1), but this was only successful for $CO_2$. In addition, maize addition lifted the contribution
of ebullition towards 56% of all emitted gaseous C compared to 37% without maize. Because of that, the estimated $\delta^{13}C\text{-}CO_{2\,|\,SOC}$ and $\delta^{13}C\text{-}CH_{4\,|\,SOC}$ used in Eq. 3 were perhaps not representative for the maize-amended pots, since the proportion of ebullition transport was particularly large in our experiment. However, decreasing ebullition by retarding mineralisation, e.g. by lowering soil temperatures or by working with a less labile substrate than the finely ground maize, would after all have but a limited impact. To improve source partitioning of emissions, the use of exogenous OC with a more strongly contrasting $\delta^{13}C$ would help the most, as then the relative effect of isotopic fractionation declines.

**5 Conclusion**

Using a stable isotope approach, it was confirmed that the addition of high-quality OC like maize shoots stimulates Fe reduction and dissolution of native SOC. The synchrony of both processes suggests that priming of SOC dissolution after addition of maize could result from increased net co-release of OC during reductive dissolution of Fe hydroxides, next to enhanced fermentation of native SOC to DOC. In support of this hypothesis, the pool of Fe-bound C ($1.3 - 2.0$ g C kg$^{-1}$), estimated by means of reduction with $NH_2OH.HCl$, was large enough to explain this. Moreover, we found that maize addition stimulated SOC dissolution more strongly in the soil with the highest SOC:Fe$_{ox}$ ratio and Fe-bound SOC content. However, positive priming of SOC mineralisation into $CO_2$ and $CH_4$ only occurred under CF in one out of the two investigated soils. Priming of SOC dissolution therefore does not necessarily result in proportional priming of native SOC-derived emissions, most likely owing to an overall preference of microbes for maize-derived DOC in anaerobic conditions. However, it turned out that it is particularly difficult to assess priming effects on native SOC mineralisation in paddy soils, because source partitioning of C emissions based on a stable isotope mass balance is inherently complicated due to various isotopic fractionation effects. Nevertheless, our study indicated that in particular in soil with a high SOC:Fe$_{ox}$ ratio, the addition of labile OC can result in substantial extra dissolution and mineralisation of native SOC under CF irrigation, with a possible adverse impact on the SOC balance in the longer term. The adoption of water-saving irrigation instead successfully decreases the stimulation of SOC dissolution caused by OC addition, and inhibits positive priming of SOC mineralisation.

**Declaration of competing interest**

The authors declare that they have no conflict of interest.

**Acknowledgements**

We are grateful for the financial support by the FACCE-JPI project "Greenhouse gas emissions from paddy rice soils under alternative irrigation management" (GreenRice). Furthermore, we would like to genuinely thank the involved staff in the



tropical greenhouse, the lab of the Soil Fertility and Nutrient Management (SoFer) group and the Isotope Bioscience Laboratory (ISOFYS) from Ghent University (Belgium).

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
