# Peer review of "Effect of organic carbon addition on paddy soil organic carbon decomposition under different irrigation regimes"

_Biogeosciences, 2021_

## Referee Comment (RC2)

This paper is dealing with the role of reductive desolution of Fe (oxyhydr)oxides in a six-week pot experiment with rice plants coupling with reduction of native organic carbon (OC) as an alternative soil electron acceptor. The goal of this study was to point out the role of Fe content on crop residue (Maize shoot), which can be fastly consumed by microorganisms or eventually stabilised as Fe-organic carbon complexes. The authors used two Bangladeshi soils with contrasting SOC-to-reducible-Fe (SOC:Feox) ratios were kept under a regime of alternate wetting and drying (AWD) or continuous flooding (CF). The topic is well-suited for a publication in Biogeosciences, but it may be improved, before publication. I suggest minor revision.

- Lines 55: In the introduction, the choice and importance of maize shoots, which can be rapidly metabolized by micro-organisms, should be discussed more largely than in lines 55
- Line 55,85: There are much older references, such as Ponnamperuma, 1984, Better find more recent references…

- How did the authors adjust and maintain the water content for Control and AWD conditions?

- The data are looked at from a maize straw addition point of view, but the initial carbon (TC) is not really looked at, or it has not been clearly explained: what is the initial carbon between substrate C, basal soil dervied C.
- What are the international standards used in lab? What are their $\delta^{13}C$ values? This should be given. These terms should be defined early in the M&M section, related to analytical procedure. This will help the reader for a better understanding.
- Lines: 90-95: Give more details for how each soil was sampled. For example, were sterilized tools used? Was just one location sampled for each soil, or a few different locations that were then bulked?

- Lines: 90-95: Moreover, why 2 different soils are taken for the experiment, with two different TC amount? It can be difficult to appreciate the difference since the initial soils are not the same for the experiment?
- Line 90-95: It was mentioned that soils were sampled 2014. Did authors check the soil physicochemical and biological properties before starting the experiment at 2018? If so how was the variations/changes of soil properties?
- Line 160-165: Why did authors used NH2OH.HCl extraction procedure to measure Fe bound C contents instead of DCB extraction procedure?

- Table 1: Better to define $Fe_{ox}$ and SOC:$Fe_{ox}$ under the table.

---

## Author Response (AR1)

**Author's response**

The editor advised general 'major revisions' of the manuscript, and therefore we have implemented all changes that we suggested in the open discussion in response to both referees. In addition, we corrected some small typos (Line 32, Line 365 and Line 495 of the revised manuscript) and updated one citation (Line 423 of the revised manuscript). The below list of point-by-point changes thus bundles and repeats these responses, with line numbers referring to the original version of the manuscript. Changes are also visible in the track-changes revised manuscript.

**Referee 1**

This manuscript investigates how the addition of exogenous OC influences dissolution and mineralization of native SOC in paddy soils in function of water management, with particular attention to the role of the co-release of Fe-bound SOC. The present data are interesting and novel, and the experiments are well designed to provide the useful insights in the complex priming effect in soil.

Response: Thank you very much for your appreciation and very valuable suggestions to improve the manuscript!

Please find some more detailed comments below.

1. Lines 17, 87, the full name of "Feox" is inconsistent. I recommend that the "oxalate-extractable Fe" is more accurate than "reducible-Fe".

   Response: We agree that there is no one-on-one match between oxalate extractable Fe and reducible Fe: We therefore would replace "reducible" by "oxalate-extractable" in Line 17.

2. Lines 81-83, it is suggested to add the Fe reduction derived production of hydroxyl radicals in the Discussion section. During this process, the production of hydroxyl radicals is certainly essential in the priming effect of native OC mineralization.

   Response: Thank you for this very interesting addition. Although we have not measured levels of hydrogen peroxide ($H_2O_2$) or hydroxyl radicals, the production of hydroxyl radicals may certainly contribute to anaerobic decomposition. However, we doubt that Fenton chemistry would have functioned as major mechanism leading to positively primed emissions upon enhanced Fe reduction. In the first place, observed priming effect coefficients for gaseous C emissions were lower than for dissolution (and were in fact often negative) (Fig. 8). This suggests that the potential contribution of the mentioned mechanism (which would mainly impact gaseous emissions, in particular $CO_2$

emissions) is rather low as compared to other avenues by which maize addition could have stimulated native SOC mineralisation. In addition, the larger share of C emitted as $CH_4$ from maize-amended pots than from control pots (Fig. 4) rather contradicts the idea that the production of hydroxyl radicals would be considerably higher in amended pots than in controls without amendment – as these would lead to relatively higher $CO_2$ emissions (not $CH_4$).

Nevertheless, we agree that it is certainly relevant to mention Fenton chemistry in Section 4.2, and we propose to add the following after Line 418.

"Lastly, enhanced Fe reduction might also abiotically mediate positive priming of native SOC mineralisation through Fenton reactions that lead to the production of reactive hydroxyl radicals (Yu & Kuzyakov, 2021). However, as priming effect coefficients for gaseous C emissions were low as compared to those for dissolution (Fig. 8), and since the relative contribution of $CO_2$ to gaseous C emissions in maize-amended pots was lower than in control pots (Fig. 4), it seems unlikely that the potentially enhanced production of hydroxyl radicals would have played a considerable role in stimulating SOC decomposition in maize-amended pots."

3.  Line 96, I think that the total Fe and dissolved Fe are critical and should be provided here.

    Response: Thank you for your suggestion. However, we tend to disagree concerning the relevance of total Fe, as the redox-active fraction of total Fe is more important than the total amount of Fe to support our objectives, so that we considered oxalate-extractable Fe the most relevant measure for Fe here (van Bodegom et al., 2003). Equivalents of dissolved Fe, furthermore, are in fact illustrated in function of time in Figure 2 (controls without maize). The Fe forms of the two soils used in this study were previously analysed by Mössbauer analysis, yielding the proportion of $Fe^{3+}$ versus $Fe^{2+}$ in various Fe-bearing minerals (Akter et al., 2018). We think that this (quite scarce) information would provide sufficient insight into potential pools of reducible $Fe^{3+}$ in the studied soils. No further changes were made to the text.

4.  Lines 156-158, why the authors did not use the typical CBD method to estimate the content of Fe-bound OC?

    Response: We could not use the citrate bicarbonate dithionite (CBD) method because this extractant contains C (in both bicarbonate and citrate). Moreover, crystalline pedogenic Fe, which is included in CBD extracts, is not very reducible (van Bodegom et al., 2003). Poorly crystalline Fe forms the most likely source of reducible Fe and is typically quantified by means of ammonium oxalate extraction. However, since oxalate again contains C, we used hydroxylamine instead, which approximately targets the

same Fe forms. We are thus convinced that hydroxylamine is a better extractant than CBD for reducible Fe in flooded soils, including for the C associated with this poorly crystalline Fe fraction. Any C that is associated with CBD-extractable crystalline Fe (if quantifiable) is also less likely to potentially contribute to enhanced dissolution of SOC, precisely because little crystalline Fe is subjected to reductive dissolution. No further changes were made to the text.

5.  Lines 366-376, root exudates can disrupt the mineral-organic associations directly or indirectly by driven redox-active bacterial communities, which are the predominant control over soil C dissolution. In the Discussion, this point should be considered.

    Response: Thank you for this pertinent remark. We propose to make the following addition in Line 373: "Next to the direct contribution of rhizodeposition, some root photosynthates (e.g. oxalate and citric acid) can also indirectly increase DOC levels by promoting the release of Fe-bound SOC in the rhizosphere through their strong metal-complexing capacity (Keiluweit et al., 2015; Yu et al., 2017). This mechanism was, however, likewise restricted here considering its local impact and the juvenile age of the rice plants."

6.  Line 63, 505, "Fe hydroxides"; Line 74, 88, "Fe3+ oxides"; Line 88, "Fe oxides"; Line 364, "Fe oxyhydroxides". The names of Fe minerals are very complexed. In fact, "Fe (oxyhydr)oxides" is more common than the above names.

    Response: We totally agree that our naming of $Fe^{3+}$ was inconsistent. Since the term "Fe oxides" encompasses both Fe oxides (e.g. hematite), Fe hydroxides (e.g. ferrihydrite) and Fe oxyhydroxides (e.g. goethite), we propose to use the term "(pedogenic) Fe oxides" throughout the manuscript, and we would clearly introduce its interpretation/definition throughout the manuscript in Line 63, where the term "Fe hydroxides" would be replaced by "Pedogenic $Fe^{3+}$ oxides, hydroxides and oxyhydroxides (hereafter collectively referred to with "Fe oxides")". We would then replace any further reference to Fe oxides in the manuscript with the term "Fe oxides".

**Referee 2**

This paper is dealing with the role of reductive desolution of Fe (oxyhydr)oxides in a six-week pot experiment with rice plants coupling with reduction of native organic carbon (OC) as an alternative soil electron acceptor. The goal of this study was to point out the role of Fe content on crop residue (Maize shoots), which can be fastly consumed by microorganisms or eventually stabilised as Fe-organic carbon complexes. The authors used two Bangladeshi soils with contrasting SOC-to-reducible-Fe (SOC:Feox) ratios were kept under a regime of alternate

wetting and drying (AWD) or continuous flooding (CF). The topic is well-suited for a publication in Biogeosciences, but it may be improved, before publication. I suggest minor revision.

Response: Thank you very much for your appreciation, as well as for your interesting suggestions!

- Lines 55: In the introduction, the choice and importance of maize shoots, which can be rapidly metabolized by micro-organisms, should be discussed more largely than in lines 55

  Response: As explained in Lines 83-84, we have specifically chosen for maize shoots because of their contrasting $δ^{13}C$ compared to native SOC. Rice-maize cropping systems are moreover common in Bangladesh and elsewhere, so incorporation of maize shoots into a field with young rice plants forms a realistic situation. We are aware that maize shoots are easily degradable by microorganisms, and regard this as an additional advantage when studying priming effects, because any stimulating or repressive effect of maize shoot addition on SOC mineralisation should relatively rapidly occur. However, the latter was not the main reason for our choice for maize shoots, so we do not consider it relevant to mention this as such in the manuscript.

- Line 55,85: There are much older references, such as Ponnamperuma, 1984, Better find more recent references…

  Response: Thank you, we agree with the remark that some references are old, and we propose to support those with additional citations of some more recent articles. In particular, in Line 53, we propose to add a reference to the article of Mandal et al. (2004), and in Line 88, we would additionally cite the paper of van Bodegom et al. (2003).

- How did the authors adjust and maintain the water content for Control and AWD conditions?

  Response: As described in a paragraph in Lines 119-125, we evaluated the water content in all pots every one or two days, and added demineralised water when necessary. For CF, we topped up the standing water each time until a mark of 2.5 cm above the soil surface. For AWD, we observed the water table by means of a perforated tube (a so-called "pani pipe"), and reflooded the pots until a mark of 2 cm above the soil surface as soon as the water table dropped more than 8 cm below the soil surface. No further changes were made to the text.

- The data are looked at from a maize straw addition point of view, but the initial carbon (TC) is not really looked at, or it has not been clearly explained: what is the initial carbon between substrate C, basal soil dervied C.

Response: We are not sure if we correctly understand this remark as intended by the referee. We suspect that the referee inquires about the C content of the soils. As described in Lines 111-112, the C content of the used maize shoots was 474.4 g C kg$^{-1}$, and Table 1 mentions that the C content of the two soils were 14.1 g C kg$^{-1}$ (Balina) and 22.4 g C kg$^{-1}$ (Sonatala). No further changes were made to the text.

- What are the international standards used in lab? What are their $\delta^{13}C$ values? This should be given. These terms should be defined early in the M&M section, related to analytical procedure. This will help the reader for a better understanding.

  Response: Thank you, we agree that it would be more transparent to mention the $\delta^{13}C$ values of standards used for calibration of the CRDS. We propose to mention in the manuscript (Line 182) that "The CRDS was calibrated using standard gases with known $\delta^{13}C$ values of -35.95 ‰ and -26.43 ‰".

- Lines: 90-95: Give more details for how each soil was sampled. For example, were sterilized tools used? Was just one location sampled for each soil, or a few different locations that were then bulked?

  Response: One field was sampled for Balina, and one for Sonatala. Within each field, 15 locations were sampled (using a common, unsterilised yet clean spade) and bulked. We propose to change the sentence "The puddle layer (0 – 15 cm) was sampled at 15 locations per field in May 2014, and stored in air-dried, ground and sieved form." (Lines 93-94) to the sentence: "Per soil series, the puddle layer soil (0 – 15 cm) of 15 locations within one field was sampled and then bulked by means of a clean spade in May 2014, after which the soil was stored in air-dried, ground and sieved form."

- Lines: 90-95: Moreover, why 2 different soils are taken for the experiment, with two different TC amount? It can be difficult to appreciate the difference since the initial soils are not the same for the experiment?

  Response: To improve the representativeness of the outcomes of the study, we included two soils with varying SOC:Fe$_{ox}$ ratio, since we mainly expected that our findings would vary in function of this parameter. However, for sure the difference between both soils is not only limited to the contents of SOC and oxalate-extractable Fe, and the variation in outcomes between both soils therefore undoubtedly also reflects differences in other parameters like the quality of SOC and the soil texture and mineralogy, in spite of our efforts to select soils with otherwise relatively limited variation. Considering the observed differences in SOC decomposition upon exogenous OC addition between the two soils, it would be very relevant to expand this experiment to other soil types, so that the results can be mechanistically linked to the soil type, and the observed dynamics could be incorporated

in biogeochemical models. We already discussed the effect of soil type in the paragraph of Lines 453-470 and did not further elaborate on this.

- Line 90-95: It was mentioned that soils were sampled 2014. Did authors check the soil physicochemical and biological properties before starting the experiment at 2018? If so how was the variations/changes of soil properties?

    Response: The initial soil properties after soil sampling were first assessed by Akter et al. (2018) in 2014, and we repeated analysis of the SOC content (and its $\delta^{13}C$) before starting the experiment in 2018. Over the course of those four years, the SOC content of both soils slightly decreased (i.e. Balina: 16.5 g C kg$^{-1}$ → 14.1 g C kg$^{-1}$ | Sonatala: 23.6 g C kg$^{-1}$ → 22.4 g C kg$^{-1}$) in spite of being stored in dry form in a cool place. We did not repeat the other mentioned analyses before starting the experiment in 2018. No further changes were made to the text.

- Line 160-165: Why did authors used NH2OH.HCl extraction procedure to measure Fe bound C contents instead of DCB extraction procedure?

    Response: We here provide the same response as to the question of referee 1 on the use of hydroxylamine: "We could not use the citrate bicarbonate dithionite (CBD) method because this extractant contains C (in both bicarbonate and citrate). Moreover, crystalline pedogenic Fe, which is included in CBD extracts, is not very reducible (van Bodegom et al., 2003). Poorly crystalline Fe forms the most likely source of reducible Fe and is typically quantified by means of ammonium oxalate extraction. However, since oxalate again contains C, we used hydroxylamine instead, which approximately targets the same Fe forms. We are thus convinced that hydroxylamine is a better extractant than CBD for reducible Fe in flooded soils, including for the C associated with this poorly crystalline Fe fraction. Any C that is associated with CBD-extractable crystalline Fe (if quantifiable) is also less likely to potentially contribute to enhanced dissolution of SOC, precisely because little crystalline Fe is subjected to reductive dissolution." No further changes were made to the text.

- Table 1: Better to define Feox and SOC:Feox under the table.

    Response: We agree that it is clearer to define $Fe_{ox}$ and SOC:$Fe_{ox}$ below Table 1, and propose to do that.

References:

Akter, M., Deroo, H., De Grave, E., Van Alboom, T., Kader, M. A., Pierreux, S., Begum, M. A., Boeckx, P. & Sleutel, S. (2018). Link between paddy soil mineral nitrogen release and iron and

manganese reduction examined in a rice pot growth experiment. *Geoderma, 326*, 9-21. doi:https://doi.org/10.1016/j.geoderma.2018.04.002

Keiluweit, M., Bougoure, J. J., Nico, P. S., Pett-Ridge, J., Weber, P. K. & Kleber, M. (2015). Mineral protection of soil carbon counteracted by root exudates. *Nature Climate Change, 5*(6), 588-595. doi:10.1038/nclimate2580

Mandal, K. G., Misra, A. K., Hati, K. M., Bandyopadhyay, K. K., Ghosh, P. K. & Mohanty, M. (2004). Rice residue-management options and effects on soil properties and crop productivity. *Journal of Food Agriculture and Environment, 2*, 224-231.

van Bodegom, P. M., van Reeven, J. & Denier van der Gon, H. A. C. (2003). Prediction of reducible soil iron content from iron extraction data. *Biogeochemistry, 64*(2), 231-245. doi:10.1023/A:1024935107543

Yu, G.-H. & Kuzyakov, Y. (2021). Fenton chemistry and reactive oxygen species in soil: Abiotic mechanisms of biotic processes, controls and consequences for carbon and nutrient cycling. *Earth-Science Reviews, 214*, 103525. doi:https://doi.org/10.1016/j.earscirev.2021.103525

Yu, G., Xiao, J., Hu, S., Polizzotto, M. L., Zhao, F., McGrath, S. P., Li, H., Ran, W. & Shen, Q. (2017). Mineral availability as a key regulator of soil carbon storage. *Environmental Science & Technology, 51*(9), 4960-4969.

---

## Referee Report (RR1)

The authors have done a nice job revising the manuscript and I have only few more suggestion as follows.

Abstract Line 25: Mind the space between "above-mentioned".

Introduction Line 40: Write the full word for SOC "Soil organic matter"

Line 610: Change the title of research paper into sentence case "Quantifying Rhizosphere Respiration in a Corn Crop under Field Conditions"

Line 615: Use superscript for "$Fe^{2+}$ and $Mn^{2+}$ "

Line 625, 645: Use subscript for "$CH_4$" and [$CO_2$]

Line 655,660: Use subscript for "$CH_4$ and $CO_2$"

Line 660: Correct the reference starting with "Zhang, X.-h., Li, L.-q., and Pan, G.-x.: "

---

## Author Response (AR2)

**Author's response**

The editor advised some minor corrections of the manuscript, based on the following suggestions of Reviewer 2.:

Abstract Line 25: Mind the space between "above-mentioned".

Introduction Line 40: Write the full word for SOC "Soil organic matter"

Line 610: Change the title of research paper into sentence case "Quantifying Rhizosphere Respiration in a Corn Crop under Field Conditions"

Line 615: Use superscript for "Fe2+ and Mn2+ "

Line 625, 645: Use subscript for "CH4" and [CO 2]

Line 655,660: Use subscript for "CH 4 and CO 2"

Line 660: Correct the reference starting with "Zhang, X.-h., Li, L.-q., and Pan, G.-x.: "

Thank you for suggesting these corrections. We have implemented all the changes that were suggested by Reviewer 2. Note that in the version of the revised manuscript with tracked changes, changes are not tracked in the "References" section owing to the use of EndNote. If marking of the changes in the "References" section would be desired, we could obviously provide this.